# Risk Factors for Anaphylaxis in Children Allergic to Peanuts

**DOI:** 10.3390/medicina59061037

**Published:** 2023-05-28

**Authors:** Tadej Petek, Mija Lajhar, Blažka Krašovec, Matjaž Homšak, Maja Kavalar, Peter Korošec, Brigita Koren, Maja Tomazin, Tina Hojnik, Vojko Berce

**Affiliations:** 1Department of Paediatrics, University Medical Centre Maribor, Ljubljanska ulica 5, 2000 Maribor, Slovenia; tadej.petek@ukc-mb.si (T.P.); brigita.koren@ukc-mb.si (B.K.); maja.tomazin@ukc-mb.si (M.T.); tina.hojnik@ukc-mb.si (T.H.); 2Faculty of Medicine, University of Maribor, Taborska ulica 8, 2000 Maribor, Slovenia; mija.lajhar@student.um.si (M.L.); blazka.krasovec@student.um.si (B.K.); 3Pediatric Outpatient Clinic, Maistrova ulica 22, 2230 Lenart v Slovenskih Goricah, Slovenia; matjaz_homsak@hotmail.com; 4Pediatric Allergy Outpatient Clinic, Lavričeva ulica 1, 2000 Maribor, Slovenia; maja.kavalar@triera.net; 5Laboratory for Clinical Immunology and Molecular Genetics, University Clinic of Respiratory and Allergic Diseases Golnik, 4204 Golnik, Slovenia; peter.korosec@klinika-golnik.si

**Keywords:** peanut allergy, children, anaphylaxis, allergen components, severity

## Abstract

*Background and Objectives*: A peanut allergy is the most common single cause of anaphylaxis in children. The risk factors for anaphylaxis in children with a peanut allergy are not well defined. Therefore, we aimed to identify epidemiological, clinical, and laboratory characteristics of children with a peanut allergy that may predict the severity of the allergic reaction and anaphylaxis. *Materials and Methods*: We conducted a cross-sectional study and included 94 children with a peanut allergy. Allergy testing was performed, including skin prick testing and the determination of specific IgE levels to peanuts and their Ara h2 component. In case of discordance between patient history and allergy testing, an oral food challenge with peanuts was performed. *Results*: Anaphylaxis and moderate and mild reactions to peanuts occurred in 33 (35.1%), 30 (31.9%), and 31 (33.0%) patients, respectively. The severity of the allergic reaction was only weakly correlated (*p* = 0.04) with the amount of peanuts consumed. The median number of allergic reactions to peanuts was 2 in children with anaphylaxis compared to 1 in other patients (*p* = 0.04). The median level of specific IgE to Ara h2 was 5.3 IU/mL in children with anaphylaxis compared to 0.6 IU/mL and 10.3 IU/mL in children with mild and moderate peanut allergies (*p* = 0.06). The optimal cutoff for distinguishing between anaphylaxis and a less severe allergic reaction to peanuts was a specific IgE Ara h2 level of 0.92 IU/mL with 90% sensitivity and 47.5% specificity for predicting anaphylaxis (*p* = 0.04). *Conclusions*: Epidemiological and clinical characteristics of the patient cannot predict the severity of the allergic reaction to peanuts in children. Even standard allergy testing, including component diagnostics, is a relatively poor predictor of the severity of an allergic reaction to peanuts. Therefore, more accurate predictive models, including new diagnostic tools, are needed to reduce the need for oral food challenge in most patients.

## 1. Introduction

The prevalence of peanut allergies in the developed world has increased significantly over the last few decades and is now estimated to be 1% to 3% of children. Peanuts are therefore one of the most common food allergens in children, and allergic reactions to peanuts are often more severe compared to other common food allergens. In addition, a peanut allergy is often lifelong and represents a major daily burden that negatively affects quality of life [1,2]. Compared to other food allergens, a peanut allergy is associated with a higher rate of accidental exposure and anaphylaxis, the latter occurring in up to 50% of all allergic reactions to peanuts in children [3].

The etiopathogenesis of peanut allergies is not fully understood, although genetic factors play an important role, and peanut allergies are much more common in children with a close family member who is allergic to peanuts [4,5]. Other risk factors for the development of peanut allergies are hypersensitivity to several other food and/or airborne allergens, chicken egg allergy, vitamin D deficiency, impaired skin barrier in patients with atopic dermatitis, and late introduction of peanuts [6,7,8,9,10].

Tolerance is more common than a peanut allergy in peanut-sensitized patients. However, the prevalence of sensitization, defined by a positive skin prick test (SPT) or the detection of specific immunoglobulin class E (sIgE) to whole peanut extract in the general population of children, is estimated to be approximately 10%. The prevalence of sensitization to peanuts is even higher in patients with atopic dermatitis or allergic respiratory disease, mainly due to sensitization to non-specific and cross-reactive peanut proteins, such as Ara h8, resulting in mild symptoms such as oral allergy syndrome (OAS) or even a tolerance to peanuts [11,12]. Therefore, patient history is very important in the diagnosis of peanut allergies [1].

The spectrum of allergic reactions to peanuts ranges from mild local reactions (such as OAS) to generalized skin reactions (e.g., urticaria), angioedema, and life-threatening anaphylaxis [13].

The potential risk factors for severe allergic reactions in peanut allergic individuals are not fully elucidated. In the adult population, an association between severe reactions to peanuts and the female sex, atopic dermatitis, a family history of atopy or peanut allergy, and allergies to house dust mites or birch pollen has been observed [14].

When an oral food challenge (OFC) was performed to diagnose food allergies in children, the severity of the peanut allergy was significantly associated with the amount of peanuts ingested and levels of sIgE to peanut. A weak association was also found for allergic rhinoconjunctivitis and the maternal history of asthma. In the same study, older age was reported as a risk factor for the severity of an accidental allergic reaction to peanuts [15]. Asthma has also been reported as an independent risk factor for anaphylaxis in food-allergic children [16].

In the evaluation of a child with a suspected peanut allergy, skin prick tests (SPT) and peanut sIgE determination are usually performed as first-line investigations. However, both tests have low specificity (while being highly sensitive) and do not correlate well with the severity of the allergic reaction [17].

Component-resolved diagnostics determine the serum IgE antibody levels against specific allergenic peanut protein components. In particular, sIgE against the Ara h2 peanut protein has been shown to be a predictive factor for allergy, and its serum level correlates with the likelihood of clinically relevant allergy and a positive OFC [4,18].

Ara h2 determination proved superior to whole peanut extract-based serology or SPT testing for the diagnosis of peanut allergies in infancy [19]. Ara h6 shares multiple epitopes with Ara h2 and also provides appropriate diagnostic accuracy, despite the latter being the dominant peanut allergen [20]. Recently, the basophil activation test (BAT) has been shown to improve the specificity of peanut allergy testing. BAT can predict the dose–response relationship when eating peanuts and thus potentially reduce the need for OFC [21,22].

The aim of our study was to identify epidemiological, clinical, and laboratory characteristics of children that may predict the severity of allergic reactions in patients with a peanut allergy.

We hypothesized that some risk factors for the development of a peanut allergy and biomarkers for its diagnosis are also associated with the severity of the peanut allergy. Therefore, we expected that more severe allergic reactions to peanuts would occur in patients with a family history of food (especially peanut) allergies, a family history of allergies to airborne allergens, a shorter duration of breastfeeding, the late introduction of peanuts and older age at first reaction, hypersensitivity to other foods or airborne allergens, and comorbidities, such as atopic dermatitis or allergic respiratory diseases. We hypothesized that children with higher allergy test values (SPT, sIgE) would have more severe allergic reactions to peanuts.

## 2. Materials and Methods

### 2.1. Participants

We performed a cross-sectional study and included all patients with a peanut allergy aged 3 months to 18 years referred to the pediatric allergy clinic of our pediatric department from 1 January 2020 to 31 December 2022. Diagnosis was based on the history of an allergic reaction within 2 h of peanut consumption and a positive allergy test. In patients with a positive history and a negative allergy test, we performed an open OFC with peanuts, and if it was negative, we did not include the patient in the study [23,24]. We also did not include patients with positive allergy tests and no reaction to peanuts, or those who had never eaten peanuts. When there was no history of anaphylaxis and the sIgE on Ara h2 was <2 IU/mL, we also performed OFC if the children and their guardians (in children aged <15 years) agreed to it (24). OFC was performed in 44 (46%) of the children included in the study.

Epidemiological data such as age, sex, family history of allergy (in parents or siblings) to food or airborne allergens, duration of breastfeeding, age at first peanut introduction, and age at first allergic reaction to peanuts were recorded. Comorbidities (atopic dermatitis, viral-induced wheeze, allergic airway disease) and allergies to other foods or airborne allergens were also recorded. The amount of peanuts that triggered an allergic reaction were also recorded, with the assumption that one peanut kernel weighs around 0.5 g [24]. We also recorded the prescription of epinephrine autoinjector before the first visit to our outpatient clinic.

Allergic reaction to peanuts was assessed according to the Ring and Messmer scale, as shown in Table 1. Reaction grading was always according to the most severe symptom in each patient [25].

When a patient had more than one allergic reaction to peanuts in his/her lifetime, the most severe, including the reaction during an OFC, was recorded for further statistical analysis. Patients were classified into three groups for statistical analysis. In the first group, patients with a mild reaction to peanuts, such as OAS or the worsening of atopic dermatitis within 2 h after peanut ingestion, were classified. Patients with urticaria and/or angioedema (grade 1 according to the Ring and Messmer scale) were classified into the second group, and patients with anaphylaxis to peanuts (grade 2 or more according to the Ring and Messmer scale) were classified into the third group.

### 2.2. Methods

Skin prick tests were performed on the volar side of the forearm using a positive control (histamine dihydrochloride 10 mg/mL), a negative control (diluent solution), and a potential allergen (Lofarma SpA, Milan, Italy). Results were recorded after 15 min, and the test was positive if the maximum diameter of the wheal was at least 3 mm larger than the negative control. Serum sIgE was measured using the Pharmacia CAP method (Uppsala, Sweden), and a cut-off value of 0.35 IU/mL was used to confirm sensitization.

An open OFC with peanuts was performed when appropriate (see above under section Participants) and always in a hospital setting. We started with 0.25 g of 100% peanut butter (or half of one peanut kernel) and doubled the dose every 30 min until signs of a type I hypersensitivity reaction appeared or the final dose was reached (4 g of peanuts in children <5 years and 8 g in children >5 years) [24,26].

### 2.3. Ethical Approval

The study was approved by the Ethics Committee of the University Clinical Centre Maribor (UKC-MB-KME-25/20) and was conducted in accordance with the Helsinki Declaration of 1975, as revised in Edinburgh in 2000. All participants or their legal guardians (for children under 16 years of age) signed an informed consent form, and additional separate consent was obtained prior to the performance of OFC.

### 2.4. Statistical Analysis

Statistical analysis was performed using IBM SPSS 24.0 software (IBM Inc. Armonk, NY, USA). The association of categorical variables (e.g., gender, family history of allergy, presence of comorbidities, allergy to other foods or airborne allergens) with the severity of the allergic reaction (considered as an ordinal variable) to peanuts was analyzed by Mann-Whitney-U test. Chi-square or Fisher exact tests were used for analyzing the association of categorial variables with anaphylaxis (as opposed to less severe reaction). The association of continuous variables (e.g., duration of breastfeeding, age at first introduction of peanut, age at first allergic reaction to peanut, amount of peanuts triggering the reaction, number of allergens to which the patient is sensitized, wheal size at SPT with peanuts, blood levels of sIgE to whole peanuts and their Ara h2 protein) with the severity of the allergic reaction was analyzed by ordinal regression after the Kolmogorov-Smirnov normality test. Receiver-operating characteristic (ROC) curve analysis was used to estimate optimal cut-off values to discriminate between anaphylaxis and less severe allergic reactions to peanuts for those variables found to be significantly associated with reaction severity. The α level for all tests was set at 0.05, and *p* values are presented for two-sided tests.

## 3. Results

### Epidemiological, Clinical, and Laboratory Characteristics and the Severity of Allergic Reaction

We enrolled 94 peanut-allergic children, 30 (31.9%) of whom were female. All of our patients were of Caucasian origin. Mild reactions to peanuts (e.g., OAS or exacerbation of atopic dermatitis) occurred in 31 (33.0%) patients, moderate reactions (urticaria and/or angioedema) in 30 (31.9%) patients, and anaphylaxis in 33 (35.1%) patients. The epidemiological, clinical, and laboratory characteristics of the patients are presented in Table 2.

More than a third (35.1%) of patients experienced anaphylaxis to peanuts, which was life-threatening in 1 (1.6%) individual. Caregivers introduced peanuts late, at a median age of 12 months, and the first allergic reaction was reported at a median age of 30 months. In most cases, it occurred only on one occasion. Regarding allergy testing, sensitization to a median of 1.5 food allergens (other than peanuts) was found, with a median SPT wheal size to peanuts, specific IgE level, and Ara h2 level of 6 mm, 8.4 IU/mL, and 4.9 IU/mL, respectively.

A comparison of the epidemiological, clinical, and laboratory characteristics of patients with a mild allergy, urticaria and/or angioedema, and anaphylaxis after peanut consumption is presented in the Table 3.

The amount of peanuts that triggered an allergic reaction was highest (median 2.5 g) in patients with a mild reaction and lowest (median 0.3 g) in patients with a moderate reaction. In patients with anaphylaxis, the amount of peanuts that triggered a reaction was between these two values, with a median of 1 g. Patients with anaphylaxis reported significantly more allergic reactions to peanuts (median two reactions) compared to non-anaphylactic patients (median one reaction).

When children with mild and moderate reactions to peanuts were combined into one group, the median sIgE level on Ara h2 in this group was 1.51 IU/mL, which was significantly lower than the group of patients experiencing anaphylaxis, with a median sIgE level of Ara h2 of 5.3 IU/mL. We also analyzed the association or correlation of all the characteristics presented in Table 3 with the degree of anaphylaxis, but no significant results were obtained.

The ROC curve analysis (Figure 1) showed that the optimal cut-off to discriminate between anaphylaxis and a less severe allergic reaction to peanuts was an sIgE level to Ara h2 of 0.92 IU/mL, with a sensitivity of 90% and a specificity of 47.5% for the prediction of anaphylaxis. The area under the ROC curve was 0.63 (*p* = 0.04; 95% CI 0.52–0.75).

A total of 26 (27.7%) patients were prescribed an epinephrine autoinjector before their first visit in our outpatient clinic, 4 (12.9%) with mild allergy (OAS or exacerbation of AD), 5 (16.7%) with moderate allergy (urticaria and/or angioedema), and 17 (51.5%) with a severe allergy (anaphylaxis) to peanuts (*p* < 0.01).

## 4. Discussion

In our study, we found that the epidemiological and clinical characteristics of children with a peanut allergy are not related to the severity of the reaction and cannot predict anaphylaxis. Regarding the results of standard peanut allergy testing, our study found that only the level of specific IgE to the Ara h2 component of peanuts was weakly associated with anaphylaxis.

We found that 35% of the children who reacted to peanuts had anaphylaxis. This percentage is higher compared to a systematic review by Bassegio Conrado et al., who reported that approximately 17% of allergic reactions to peanuts in European children occurred as anaphylaxis [27]. The percentage of anaphylaxis in our patients would have been even higher if we had not included mild reactions (such as OAS or exacerbation of atopic dermatitis) to peanuts, which are often reported only by parents. However, the higher percentage of anaphylaxis observed is probably due to the design of our study, as in patients with multiple allergic reactions to peanuts, only the most severe one was considered for statistical analysis. Ethnicity may influence the prevalence of peanut sensitization in the developed world and is more common in African American children [28]. However, there are no data to suggest that ethnicity influences the severity of a peanut allergy itself. Therefore, we do not believe that the ethnic composition of our patients (all Caucasian origin) influenced our results.

Risk factors for peanut allergy, such as a family history of peanut allergies, atopic dermatitis, allergic airway disease, allergies to other foods or airborne allergens, and a late introduction of peanuts [4,5,6,7,8,9], were not associated with the severity of peanut allergy in our study.

We also did not confirm the findings of Datema et al., who reported that several epidemiological risk factors (female sex, early onset of peanut allergy, atopic dermatitis, familial atopy, and sensitization to house dust mite) were associated with a more severe allergic reaction to peanuts in peanut-allergic adults [14]. However, we confirmed their finding that the level of sIgE to the peanut protein Ara h2 is an independent risk factor for anaphylaxis in peanut-allergic adults; although, in our study, we did not find significant differences in the level of sIgE to Ara h2 between the groups with moderate and severe allergic reactions. The mean level of sIgE to Ara h2 was even higher (although not significantly) in our group with a moderate reaction to peanuts compared to patients with anaphylaxis. The optimal cut-off value of sIgE to Ara h2 to discriminate between anaphylaxis and less severe reactions to peanuts is therefore not easy to determine, and the value of 0.92 IU/mL is sensitive (90%) at the expense of less than 50% specificity.

This threshold is higher than 0.35 IU/mL, as suggested by Keet et al. and Nicolaeu et al., although they reported similar sensitivity and slightly higher specificity. The higher cut-off value in our study may be explained by different study designs, as Keet et al. and Nicolaeu et al. determined a cut-off value that would distinguish between clinically significant allergies and harmless sensitization [24,29], whereas we tried to determine a cut-off value that would distinguish between anaphylaxis and milder forms of peanut allergy. A similar study was performed by Martinet et al., who reported that sIgE levels to Ara h2 below 0.44 IU/mL were associated with a low risk of anaphylaxis in children, whereas levels above 14 IU/mL were associated with a high risk of anaphylaxis [30]. Our results agreed with those of Santos et al., who established an optimal cutoff level of sIgE to Ara h2 of 1.4 IU/mL to discriminate between severe and less severe allergic reactions to peanuts during OFC. In contrast to our results, they reported that SPT and sIgE to whole peanuts also correlated with the severity and lower threshold for the occurrence of an allergic reaction to peanuts during OFC. Santos et al. also found that the BAT test was the best predictor of the severity and threshold of allergic reaction to peanuts during OFC [22].

We also found an association between the amount of peanuts consumed and the severity of the allergic reaction and an association between a higher number of previous allergic reactions and anaphylaxis. These findings suggest that allergic reactions of different severities may occur in the same patient, depending also on the amount of peanuts consumed.

We would expect a higher proportion of studied children to be prescribed an epinephrine autoinjector after anaphylaxis, as the European Academy of Allergy and Clinical Immunology guidelines recommend the prescription of an epinephrine autoinjector in all children allergic to nuts or peanuts [31]. This low proportion of patients prescribed an epinephrine auto-injector may be explained by the fact that not all children with an allergic reaction to peanuts, or even anaphylaxis, were seen in the appropriate emergency department at the time of the reaction itself. For some, a severe allergic reaction went unrecognized. Furthermore, local emergency medical services often do not prescribe an autoinjector until the allergy is proven.

Our study has several limitations. We had to rely on the history presented by the caregivers regarding the severity of the allergic reaction in a significant proportion of the patients. Secondly, the amount of peanuts and the severity of the allergic reaction depend on the circumstances and may differ between an allergic reaction triggered during OFC and treated appropriately and an accidental and uncontrolled domestic peanut ingestion. We sought to address these issues by performing OFC in all patients with mild and moderate reactions that were not documented in the medical records or when the history of anaphylaxis presented by caregivers and the results of allergy testing were inconclusive.

## 5. Conclusions

In conclusion, epidemiological, clinical, and laboratory characteristics as risk factors for anaphylaxis are of limited value in peanut-allergic children, and reactions of different severity may occur in the same patient. Therefore, more accurate diagnostic tools that can also predict dose–response relationships (e.g., BAT) may be useful in clinical practice in selected patients. Children with a clinically manifested peanut allergy still need to be treated with extreme caution and prescribed with an epinephrine autoinjector.

## Figures and Tables

**Figure 1 medicina-59-01037-f001:**
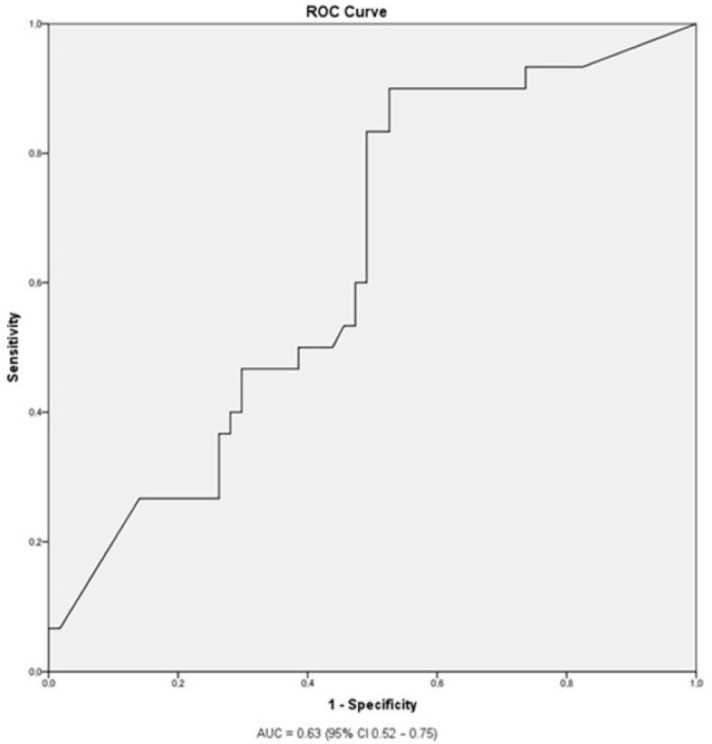
The receiver operating characteristic (ROC) curve analysis of specific IgE levels to Ara h2 discriminating between anaphylaxis and less severe peanut allergy.

**Table 1 medicina-59-01037-t001:** Ring and Messmer grading scale for allergic reactions (25).

Grade	Skin	Abdomen	Respiratory Tract	Cardiovascular System
I	Itch Flushing Urticaria Angioedema	No symptoms	No symptoms	No symptoms
II	Itch Flushing Urticaria Angioedema	Nausea Cramps	Rhinorrhea Hoarseness Dyspnea	Tachycardia (≥20 bpm rise in heart frequency) Hypotension (≥20 mm Hg fall of SBP) Arrythmia
III	Itch Flushing Urticaria Angioedema	Vomiting Defecation	Laryngeal edema (stridor) Bronchospasm Cyanosis	Circulatory shock
IV	Itch Flushing Urticaria Angioedema	Vomiting Defecation	Respiratory arrest	Circulatory arrest

Bpm, beats per minute. SBP, systolic blood pressure.

**Table 2 medicina-59-01037-t002:** Characteristics of children with peanut allergy.

Characteristic	Frequency (n)	Percentage (%)
Female sex	30	31.9
Breastfeeding *	62	67.4
Family history of food allergy	23	25
of peanut allergyFamily history of allergy to airborne allergens	15 57	16.3 60.6
Atopic dermatitis	69	75
Asthma/viral induced wheeze	33	35.9
Allergic rhinitis	33	35.9
Allergy to other foods	64	69.6
to other nuts	44	47.8
Allergy to airborne allergens	53	57.6
Anaphylaxis to peanuts (total)	33	35.1
grade 2 **	21	33.3
grade 3 **	11	17.5
grade 4 **	1	1.6
**Characteristic**	**Median**	**IQR**
Age at first introduction of peanuts (months)	12	12
Age at first reaction to peanuts (months)	30	33
Amount of peanuts (g) ***	1	2.6
Number of allergic reactions to peanuts	1	1
Number of food allergens ****	1.5	4
Skin test to peanuts (mm)	6	9
Serum total IgE levels (IU/mL)	328	729
Serum specific IgE levels to peanuts (IU/mL)	8.4	90.8
Serum specific IgE levels to Ara h2 (IU/mL)	4.9	82.7

* At least four months of breastfeeding. ** Percentage of all cases with anaphylaxis according to Ring and Messmer scale [23]. *** Amount of peanuts that triggered the allergic reaction. **** Number of food allergens (other than peanuts) to which sensitization was found. IQR, interquartile range.

**Table 3 medicina-59-01037-t003:** Association of epidemiological, clinical, and laboratory characteristics of children with peanut allergies with the severity of the allergic reaction.

Characteristic [n (%)] *	Mild Allergy ** (n = 31)	Urticaria/Angioedema (n = 30)	Anaphylaxis (n = 33)	*p*-Value	*p*-Value for Anaphylaxis (Yes/No) ***
Female sex	11 (35.5)	11 (36.7)	8 (24.2)	0.77	0.25
Breastfeeding ^◊^ Family history of allergy to airborne allergens	18 (58.1) 15 (48.4)	20 (66.7) 22 (73.3)	24 (72.7) 20 (60.6)	0.62 0.51	0.65 1.00
Family history of food allergy to peanuts	6 (19.4) 6 (19.4)	8 (26.7) 4 (13.3)	9 (27.3) 5 (15.1)	0.70 0.64	0.81 1.00
Atopic dermatitis	21 (67.7)	22 (73.3)	26 (78.8)	0.90	0.62
Asthma/viral induced wheeze	8 (25.8)	14 (46.7)	11 (33.3)	0.37	0.66
Allergic rhinitis	10 (32.3)	11 (36.7)	12 (36.4)	0.96	1.00
Allergy to other foods to other nuts	20 (64.5) 13 (41.9)	18 (60) 12 (40)	26 (78.8) 19 (57.6)	0.87 0.87	0.16 0.16
Allergy to airborne allergens	13 (41.9)	19 (63.3)	21 (63.6)	0.14	0.66
**Characteristic** **[median (IQR)]**					
Age at first introduction of peanuts (months)	12 (0)	13 (6)	13 (23)	0.06	0.53
Age at first reaction to peanuts (months)	24 (29)	18 (22)	36 (30)	0.60	0.53
Amount of peanuts (g) ^◊◊^	2.5 (7.1)	0.3 (0.6)	1 (1.9)	**0.03**	0.40
Number of allergic reactions to peanuts	1 (0)	1 (1)	2 (2)	0.43	**0.05**
Number of food allergens ^◊◊◊^	1 (4)	1 (5)	2 (4)	0.28	0.22
Skin test to peanuts (mm)	5 (9)	5 (8)	7 (10)	0.47	0.52
Serum total IgE levels (IU/mL)	441 (1005)	347 (1463)	482 (1209)	0.97	0.54
Serum specific IgE levels to peanuts (IU/mL)	2.5 (20.6)	15.7 (98.4)	9.4 (96.9)	0.13	0.10
Serum specific IgE levels to Ara h2 (IU/mL)	0.6 (8.1)	10.7 (89.1)	5.3 (98.3)	0.06	**0.04**

* Number of patients (percentage in parenthesis). ** Oral allergy syndrome (OAS) or worsening of atopic dermatitis within two hours after peanut consumption. *** Second *p*-value refers to the comparison of the characteristics of patients with anaphylaxis with the other two groups of patients pooled together. ^◊^ At least four months of breastfeeding. ^◊◊^ Amount of peanuts that triggered the allergic reaction. ^◊◊◊^ Number of food allergens (other than peanuts) to which the allergy was diagnosed.

## Data Availability

The raw data used in this study are openly available in Kaagle at https://doi.org/10.34740/kaggle/dsv/5528447.

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
