# Peer review of "Risk Factors for Anaphylaxis in Children Allergic to Peanuts"

_medicina, 2023, doi:10.3390/medicina59061037_

Round 1
Reviewer 1 Report
It was my great pleasure to review this manuscript. In this cross-sectional study, the authors examined 94 children with peanut allergy with the aim of identifying epidemiologic, clinical, and laboratory features that may predict the severity of allergic reaction and anaphylaxis.
The evaluation of epidemiologic, clinical, and laboratory characteristics in the paediatric population with peanut allergy is a good idea.
In the introduction, the authors have nicely presented the problems related to allergic reactions to peanuts in children. At the end of the Introduction section, they state the main objectives of the work and provide a clear hypothesis.
In the Materials and Methods section, the subjects and the methods they used in their study are clearly described.
The results are clearly tabulated and explained in the text portion of the Results section.
The discussion presents research that has produced similar results, but also points out the differences between their results and those of other authors and highlights the shortcomings of their research.
References are listed exactly as directed by the journal, and all relevant research in the area covered by the article is listed.
In general, the study seems to be well structured and relevant from a scientific point of view and to have a good number of patients, considering that it is a paediatric study.
Overall recommendation:
This is a very interesting article, and I recommend that you seriously consider publishing it in Medicine.

Author Response
The first reviewer had no comments.
Reviewer 2 Report
In this work, the authors explored whether that risk factors, including epidemiological, clinical and laboratory characteristics, can be used to distinguish the severity of peanut allergy in children. Overall, the experiments were carried out with care. The objective and hypothesis were clearly stated, and this paper is well-written and easy to follow. Several issues need to be addressed before publication.
1. The results in tables should be presented with a more detailed description.
2. Authors did not mention the race or country of the participants, which might influence the results. Associations between serum-specific IgE and human races have been reported. Authors can improve a little bit the discussion, by giving more critical input on how races affected the allergy-related serum metrics.
2. Some references are years old. References can be improved. Please consider the following papers.
<Ara h 2 is the dominant peanut allergen despite similarities with Ara h 6. DOI: https://doi.org/10.1016/j.jaci.2020.03.026>
< Ara h 2–specific IgE is superior to whole peanut extract–based serology or skin prick test for diagnosis of peanut allergy in infancy. DOI: https://doi.org/10.1016/j.jaci.2020.11.034>
3. Please indicated the AUC value in Figure 1.
